# Colonization times in Moran process on graphs

Lenka Kopfová[1,2], Josef Tkadlec [1*]

**1** Computer Science Institute, Charles University, Prague, Czech Republic, **2** IST Austria, Klosterneuburg, Austria

\* josef.tkadlec@iuuk.mff.cuni.cz

## Abstract

Moran Birth-death process is a standard stochastic process that is used to model natural selection in spatially structured populations. A newly occurring mutation that invades a population of residents can either fixate on the whole population or it can go extinct due to random drift. The duration of the process depends not only on the total population size $n$, but also on the spatial structure of the population. In this work, we consider the Moran process with a single type of individuals who invade and colonize an otherwise empty environment. Mathematically, this corresponds to the setting where the residents have zero reproduction rate, thus they never reproduce. The spatial structure is represented by a graph. We present two main contributions. First, in contrast to the Moran process in which residents do reproduce, we show that the colonization time is always at most a polynomial function of the population size $n$. Namely, we show that colonization always takes at most $\frac{1}{2}n^3 - \frac{1}{2}n^2$ expected steps, and for each $n$, we identify the slowest graph where it takes exactly that many steps. Moreover, we establish a stronger bound of roughly $n^{2.5}$ steps for undirected graphs and an even stronger bound of roughly $n^2$ steps for so-called regular graphs. Second, we discuss various complications that one faces when attempting to measure fixation times and colonization times in spatially structured populations, and we propose to measure the real duration of the process, rather than counting the steps of the classic Moran process.

**Data availability statement:** Code for the figures and the computational experiments is available from the Figshare repository: https://doi.org/10.6084/m9.figshare.27822720.

## Author summary

Consider an invasive species that is about to colonize an otherwise empty environment. In the absence of natural enemies, the species will eventually spread everywhere, but the time until the colonization is completed will depend on the exact spatial layout of the individual sites. In this work, we analyze this colonization time for various spatial layouts. We give precise formulas for the average colonization time for several commonly studied spatial layouts (such as well-mixed populations, cycles, or stars). For each population size $n$, we also identify the slowest layout and show that the corresponding

**Funding:** This study was supported by grants PRIMUS/24/SCI/012 and UNCE 24/SCI/008 from Charles University to JT. The funders had no role in study design, data collection and analysis, decision to publish, or preparation of the manuscript.

(slowest possible) colonization time is cubic in the population size $n$. Moreover, we prove an asymptotically tight general bound on the colonization time that applies to any lattice-like layout, and another bound that applies to any layout in which all connections are two-way. We conclude by discussing the implications of our results for further study of a related, well-researched quantity called the fixation time.

## Introduction

Natural selection is a stochastic process that acts on populations of reproducing individuals [1–4]. As time goes by, individuals acquire mutations that affect their reproductive rate. The advantageous mutations generally tend to propagate through the population, whereas the frequency of disadvantageous mutations tends to go down. When mutations are sufficiently rare, the key question is to determine the fate of a single newly occurring mutation as it attempts to invade a homogeneous population of residents. This fate depends on several factors, such as the population size $n$ or the relative fitness advantage $r$ that the mutation grants onto its bearer.

Another important factor that greatly affects the evolutionary dynamics is the spatial structure of the population [5,6]. Evolutionary graph theory is a framework developed to study those effects [7]. The individuals are represented as nodes of a graph (network), and the connections between the nodes represent the possible dispersal patterns. The connections can be one-way or two-way. Graphs can represent arbitrary spatial structures, including well-mixed populations, metapopulations [8–11], or lattices [12,13]. The evolutionary dynamics is commonly governed by the Moran Birth-death process [7,14]. That is, in each step, first an individual is selected for reproduction with probability proportional to its fitness, and then the offspring migrates and replaces a random neighbor, see Fig 1. (For comments on death-Birth updating, see S1 Text.)

Two key quantities that describe the fate of a new mutation are fixation probability and fixation time [15–18]. Fixation probability is the chance that the mutation eventually spreads throughout the population. Fixation time is the number of steps of the Moran process until this happens, and it captures the duration of the process. Both quantities have been studied extensively [19–27]. For example, it is known that certain graphs dramatically increase the fixation probability of even mildly advantageous mutations [28–30], though such a boost must always come at the cost of an increase in fixation time [31].

The fixation time crucially depends on the graph of the population. For example, when the initial mutant has a constant fitness advantage $r > 1$, the number of steps is roughly $n \log n$ for the well-mixed population [32], roughly $n^2$ for a population arranged along a cycle [33], and roughly $n^3$ for a population organized as a so-called double star [34]. The double stars are known to be essentially the slowest possible structures among those where all connections are two-way [34]. However, when some connections are one-way, the fixation time can be as large as exponential in $n$ [35]. In general, no efficient algorithm is known to compute or even approximate the fixation time on a given graph that contains one-way edges.

Given those difficulties, Moran process on population structures is often studied in different limits that make the analysis more tractable. The limit $r \to 1$ is called *weak selection*. It corresponds to settings where the mutation grants only a marginal advantage. That is, the invading mutants reproduce only barely more frequently than the existing residents. Evolutionary dynamics under weak selection can be approximated for any population structure [36–39]. Moreover, formulas for fixation times are known [40,41].

**Fig 1. Moran Birth-death process on a spatial structure. a,** The spatial structure is given as a network (graph), where nodes represent sites and arrows represent possible migration patterns. Nodes occupied by mutants are green (here $u$ and $v$). **b,** In each step of the Moran process, first, a random node is selected for reproduction, and then the offspring migrates along a random outgoing edge. Here, the offspring of $u$ migrated to $w$.

In the opposite limit, mutants reproduce at a much higher rate than the residents. This limit is called the *ecological scenario* [42]. It has been studied, for instance, in the context of death-Birth updating [43,44], in order to obtain approximations for the fixation probabilities [45], or when analyzing range expansion [46]. Mathematically, the ecological scenario corresponds to the setting in which residents have a reproductive rate 0, thus the mutant relative fitness advantage $r$ satisfies $r \to \infty$. Biologically, this regime thus models situations such as a new invasive species colonizing an initially empty spatially structured environment. In the ecological scenario, mutants eventually expand to all reachable parts of the environment. The *colonization time* is the expected number of steps until this happens. That is, colonization time is a direct analogue of the fixation time in the limit $r \to \infty$.

In this work, we consider colonization times on arbitrary graphs, with or without one-way edges. As our main theoretical result, for every population size $n$ we precisely pinpoint the unique population structure with the slowest colonization time, and we show that this slowest colonization time is of the order of $n^3$ steps. Thus, while fixation times on some graphs with fixed $r > 1$ may be exponentially long, colonization time on any graph is always at most polynomial. Moreover, we present a stronger bound of $n^{2.5}$ steps for the undirected graphs (where all connections are two-way) and an even stronger bound of $n^2$ steps for those graphs in which each node in the network has the same number of neighbors. To conclude, we discuss and compare several possible ways to measure colonization times and fixation times in spatially structured populations.

## Model

In this section, we describe the notions of our model in detail.

### Spatial structure

The population structure is represented by a directed graph $G$. The nodes of $G$ represent individuals, and the graph edges correspond to the connections between them. At any point in time, each node is either a *mutant* or a *resident*. Initially, there is only a single mutant at node $v$. We require that there exists a directed path from $v$ to any other node. This guarantees that in the limit $r \to \infty$ the process terminates with mutant fixation, with probability 1 in finite expected time. At any given time, we denote by $M$ the set of nodes that are mutants, and we refer to the tuple $(G, M)$ denoting the graph and its current set of mutants as a *state* of the process. A special case of graphs we concentrate on is the class of *undirected* graphs in which all connections are two-way. For a vertex $v$, we define its *(out)degree* denoted as $\deg(v)$ to be the number of outgoing edges incident with $v$. An even more special case is the class of *regular*

graphs in which all vertices have the same degree. This class includes, for example, lattices of any degree.

## Moran process at $r \to \infty$

We consider a modified version of the classic Moran birth-death process in which only mutants reproduce. In other words, mutants have fitness 1 and residents have fitness 0, thus the relative mutant advantage is $r \to \infty$. This process models situations in which an invading type colonizes an otherwise empty environment. In each step of this modified Moran process, we first pick a uniformly random node. If the node is a mutant, it reproduces onto a random neighbor; otherwise, nothing happens. More formally, suppose we pick a node $u$:

1. If $u$ is a mutant, we select another node $u'$ uniformly at random from among the $\deg(u)$ nodes connected to $u$, and we set $u'$ to be a mutant too. (Note that this changes the state of the population if and only if $u'$ used to be a resident.)
2. If $u$ is a resident, no change occurs.

## Colonization time

Given a graph structure $G$ and a starting node $v$, we define the colonization time $T(G, v)$ to be the expected number of steps until mutants fixate when the modified Moran Birth-death process is run on the graph $G$. Our primary goal here is to prove upper bounds on the quantity $T(G, v)$. Thus, we also denote by $T(G) = \max_{v \in V(G)} T(G, v)$ the maximum expected colonization time over all possible starting nodes $v$. Note that the colonization time accounts for all the steps of the process, including those steps in which a resident node was picked and did not reproduce. While perhaps counter-intuitive at first, this way of measuring time turns out to better correspond to the "real" duration of the process. See section Discussion, for an in-depth discussion of the connections among different ways of measuring time.

## Asymptotic notation

Throughout this text, we use the asymptotic notation $o(\cdot)$, $O(\cdot)$, $\Omega(\cdot)$ and $\Theta(\cdot)$ to denote that some function $f$ is asymptotically strictly smaller than some other function $g$ (denoted $f = o(g)$), asymptotically smaller than or equal to $g$ ($f = O(g)$), asymptotically larger than $g$ ($f = \Omega(g)$) and asymptotically equal to $g$ ($f = \Theta(g)$). When $f = \Theta(g)$ we say that $f$ is roughly equal to $g$. We will also use the symbol $\approx$ to denote "approximately equal to", meaning $f(n) \approx g(n)$ if $f(n) = g(n) + o(g(n))$. For example $\frac{1}{2}n^2 + 3n = o(n^3) = O(n^2) = \Omega(n \log n) = \Theta(n^2)$. See [47].

# Results

Our main analytical results are bounds on the colonization time for different classes of graphs. In particular, we show a general upper bound of $O(n^3)$ that applies to all graphs and we prove that it is tight. Then, we proceed by improving this upper bound to $O(n^{2.5})$ for undirected graphs and to $O(n^2)$ for regular graphs. We also compute asymptotically precise colonization times for specific graph classes such as the complete graphs $K_n$, the cycle graphs $C_n$, the star graphs $S_n$, and other graphs.

## General bounds

First, we consider the general setting with no constraints on the population structure. We prove that, in this case, the colonization time is always at most cubic in the population size $n$.

**Theorem 1** (General upper bound). *Let $G_n$ be a graph (directed or undirected) with n nodes. Then $T(G_n) \leq \frac{1}{2}n^3 - \frac{1}{2}n^2$.*

In particular, the colonization time on any population structure is at most polynomial in the population size $n$. Note that this contrasts with the regime of finite $r > 1$. In that regime, the fixation time on some graphs is known to be exponential [35].

The idea behind the proof is to decompose the process into stages such that each stage lasts until we gain a new mutant. We then argue that for any graph, each individual stage can take at most $O(n^2)$ steps on average. Since in total, there are $n{-}1$ stages, by linearity of expectation this gives a cubic upper bound for the total number of steps. See S1 Text for details.

Somewhat surprisingly, we show that the upper bound in Theorem 1 is exactly tight. That is, we identify a population structure which we call a *backward graph* $B_n$ for which the bound is achieved with equality, see Fig 2a.

**Theorem 2.** *For every n there exists a directed graph $B_n$ and an initial mutant node v of $B_n$ such that $T(B_n, v) = \frac{1}{2}n^3 - \frac{1}{2}n^2$.*

To sum up, the longest possible colonization time on any population structure is equal to $\frac{1}{2}n^3 - \frac{1}{2}n^2$. We also show that the shortest possible colonization time is of the order of at least $n \log n$ steps and that this is the case e.g. for the complete graph $K_n$. See Theorems 8 and 11 in the S1 Text for details. Finally, we show that the colonization time on a so-called total order graph $TO_n$ is of the order of $n^2$ steps, see Fig 2b.

## Stronger bound for undirected graphs

Our first result shows that the colonization times range from roughly $n \log n$ steps to roughly $n^3$ steps. Here we show that when the graph is undirected, that is, all connections are two-way, then the upper bound can be improved to roughly $n^{2.5}$ steps.

**Theorem 3.** *Let $G_n$ be an undirected graph with n nodes. Then $T(G_n) \leq 4n^2\sqrt{n} + o(n^2\sqrt{n}) = O(n^2\sqrt{n})$.*

Recall that in the regime of finite $r > 1$, the undirected graph with the largest known fixation time is the so-called double star $D_n$ [34], see Fig 3a. The fixation time is of the order of roughly $n^3$ steps. Thus, Theorem 3 shows that even in the special case of undirected graphs, the colonization times could generally be asymptotically shorter than the fixation times in the regime of finite $r > 1$.

The idea behind the proof is again to divide the process into stages. While a single stage can be relatively long, we are able to argue that any such "long" stage must be balanced off by several subsequent "short" stages. By amortization, we then prove that the stages take at most $O(n\sqrt{n})$ steps, on average. See S1 Text for details.

We also compute colonization times on several specific undirected graphs. We show that for both the star $S_n$ and the double star $D_n$ the colonization time is of the order of $n^2 \log n$ steps. In fact, we show that the colonization time on the star is a constant factor larger than the time on the double star. This is in contrast to the regime of fixed $r > 1$, where the fixation time on the star is also of the order of $n^2 \log n$ steps [25], whereas the fixation time on the double star is of the order of roughly $n^3$ steps [34]. For the cycle graph $C_n$, we show that the colonization time is of the order of $n^2$ steps, the same as its fixation time in the regime $r > 1$. See Fig 2 for an illustration and the S1 Text for details.

### Even stronger bound for regular graphs

Previous research has highlighted the importance of regular graphs, that is, graphs in which each node is connected to the same number of neighbors. Such graphs are also sometimes called isothermal graphs, since in the neutral evolution, each node is on average replaced equally often by its neighbors. The class of regular graphs includes lattice-like structures of any degree $d$. The Isothermal theorem states that the fixation probability of a single mutant with relative fitness advantage $r > 1$ is the same for all regular graphs [32].

As our final analytical result, for regular graphs we improve the upper bound on the colonization time to $O(n^2)$.

**Theorem 4.** *Let $G_n$ be a regular undirected graph with $n$ nodes. Then $T(G_n) = O(n^2)$.*

Note that the upper bound is asymptotically shorter than the colonization time on the star graph, which is of the order of $n^2 \log n$. Thus, regular undirected graphs generally have shorter colonization times than undirected graphs, which in turn have shorter colonization times than arbitrary graphs.

The idea behind the proof is similar to the proof of Theorem 3. We again split the process into stages. Since the graph is regular, we are able to argue that any single "long" stage must be followed by "many" short stages. The amortization argument then gives a stronger upper bound as compared to the case of undirected graphs. See S1 Text for details.

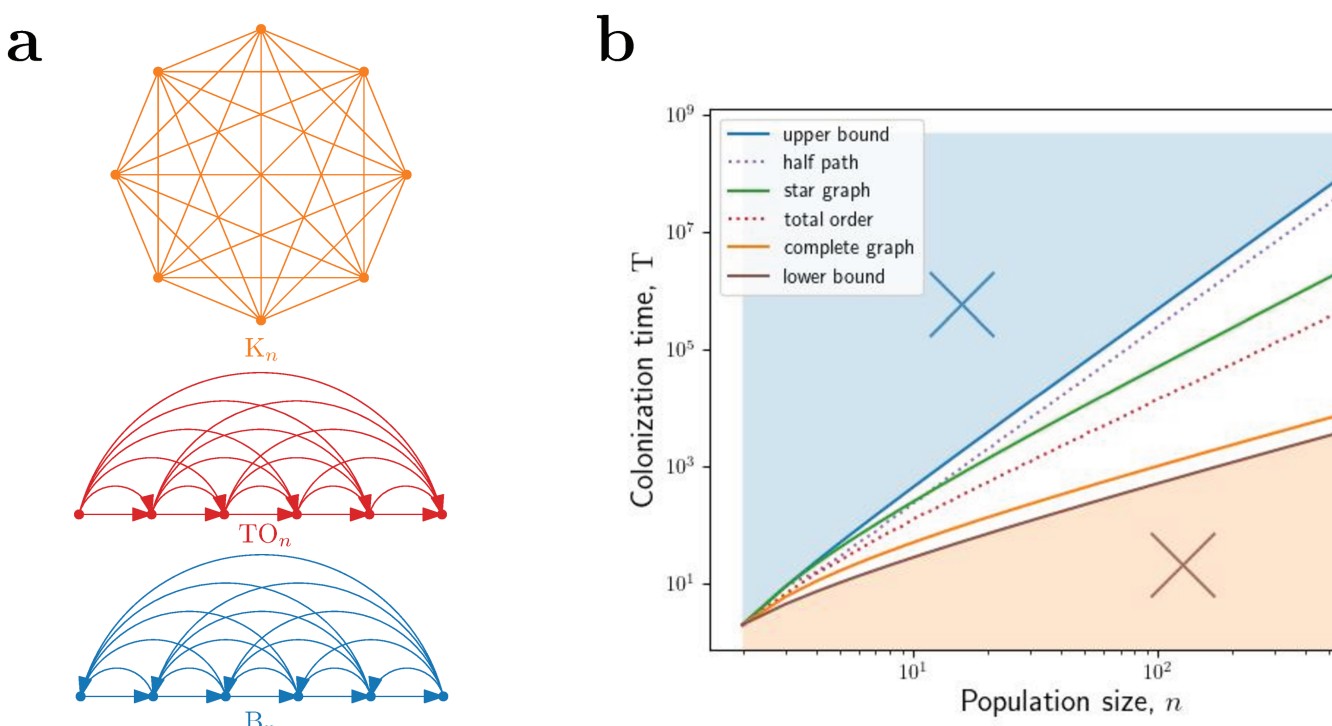

**Fig 2. Colonization times on directed graphs. a,** In the complete graph $K_n$, each two nodes are connected by a two-way edge. In the total order graph $TO_n$, the nodes are arranged left to right and all edges going left to right are included. The backward graph $B_n$ consists of a directed path going left to right, plus all one-way edges going in the opposite direction, right to left. The half-path graph is a certain variation of the backward graph in which all edges go left to right (see S1 Text for details). **b,** For each $n$, the backward graph $B_n$ (blue) is the graph with maximal colonization time. We have $T(B_n) = \frac{1}{2}n^3 - \frac{1}{2}n^2$. The shortest possible colonization time is of the order of $n \log n$ steps (see Theorems 8 and 11 in the S1 Text), which is achieved for the complete graph $K_n$ (orange). For the total order graph $TO_n$ (red) we have $T(TO_n) = \Theta(n^2)$. Here the lines show the proved analytical results, the dots show the simulations, and the axes are log-scale.

We note that the dependence on $n$ in Theorem 4 can not be improved. This is because for the cycle graph $C_n$ the colonization time is of the order of $n^2$ steps.

## Discussion

The fixation time of a newly occurring mutation is a key factor in evolutionary dynamics. Apart from depending on the population size $n$, the fixation time also depends on the relative mutant fitness advantage $r$ and on the spatial structure of the population. When the mutant fitness advantage $r > 1$ is fixed, there exist large graphs for which the fixation time is exponentially large in the population size $n$ [35]. However, as we show in this work, this kind of long-term coexistence of invading mutants and existing residents can not occur in the limit of large mutant fitness advantage $r \to \infty$, which corresponds to a species colonizing an empty environment. In this regime, the colonization time on the slowest possible graph, which we call a backward graph, is only $\frac{1}{2}n^3 - \frac{1}{2}n^2$ steps.

Existing literature in the field of evolutionary graph theory highlighted the role of spatial structures in which all connections are two-way [19,23,30]. Those structures are described by undirected graphs. The slowest known undirected graphs are the so-called double stars $D_n$. For any fixed $r > 1$, the expected fixation time on a double star $D_n$ is roughly $n^3$ steps [34]. In contrast, here we show that in the regime $r \to \infty$, the expected fixation time on any undirected graph is of the order of at most $n^{2.5}$ steps. In particular, the fixation time on double stars drops

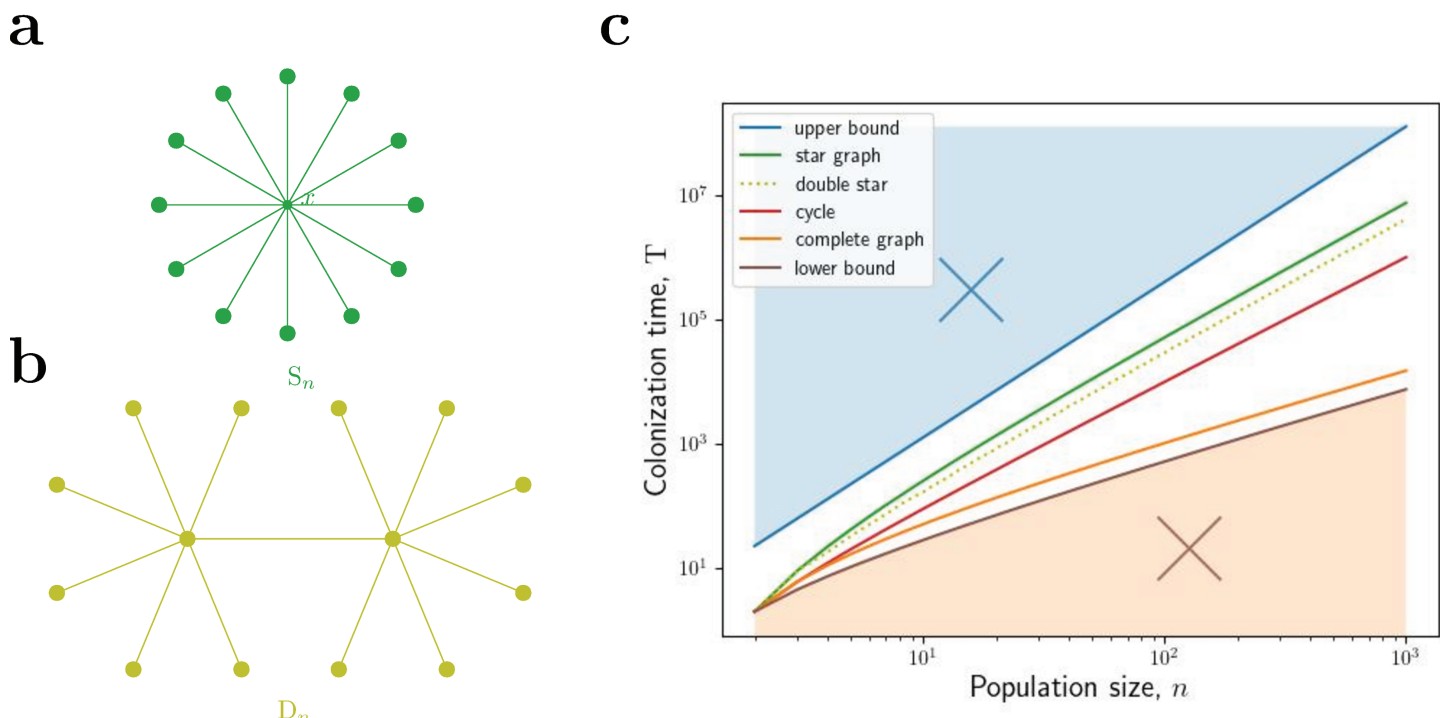

**Fig 3. Colonization times on undirected graphs. a,** In the star graph $S_n$, one node is the center, and all the other nodes are connected to it by a two-way edge. The double star graph $D_{2k}$ is obtained by joining the centers of two star graphs $S_k$ using a two-way edge. **b,** The proved upper bound for undirected graphs is $4n^2\sqrt{n} + o(n^2\sqrt{n})$. Here we plot the function $4n^2\sqrt{n}$ with blue color. The graph with the largest colonization time we found is the star graph $S_n$ (green). Again the shortest possible colonization time is of the order of $n \log n$ steps (see Theorems 8 and 11 in the S1 Text), which is achieved for the complete graph $K_n$ (orange). For the double star graph $D_n$ (yellow) we have $T(D_n) = \Theta(n^2 \log n)$. For the cycle $C_n$ (red) we have $T(C_n) = \Theta(n^2)$. Here the lines show the proved analytical results, the dots show the simulations, and the axes are log-scale.

to roughly $n^2 \log n$. Moreover, we show that double stars cease to be the slowest graphs since (plain) stars are a constant factor slower (see S1 Text for details). While stars are the slowest undirected graphs that we found, in principle there could exist undirected graphs with colonization times as long as $n^{2.5}$. Identifying the exact slowest undirected graphs is an interesting problem left for future work. We note that any such graphs, if they exist at all, would have to be irregular, since for regular graphs we proved an even stronger upper bound of at most $n^2$ steps.

Rigorous analysis of fixation times on graphs is made difficult by several factors. In what follows we elaborate on four of them.

First, the fixation time is not a number but a random variable (unlike e.g. the fixation probability). That is, depending on what individuals are selected for reproduction at each step, the total number of steps could be very small or very large. The standard approach to treat this is to study the *expected* fixation time, that is, to replace the random variable with its expectation (a number). This is often quite sufficient since the random variable is typically well concentrated [22,33]. (We note that for special graphs such as cycles or complete bipartite graphs the full distribution of the fixation time is understood [33,48].)

Second, there are in fact two competing notions of fixation time that differ in what evolutionary trajectories are taken into account. One notion is the unconditional fixation time (also known as the absorption time) which averages over all evolutionary trajectories, regardless of whether the mutants have fixated or gone extinct. Alternatively, one can consider the conditional fixation time which averages over only those trajectories in which the mutants have fixated. The two times could be quite different. For example, for a single mutant with fitness $r = 1 + \varepsilon$ who is invading a large well-mixed population the absorption time is roughly $2n \log n$ steps, whereas the conditional fixation time is roughly $\frac{2}{\varepsilon} \cdot n \log n$ steps [25,Theorem 4]. When $\varepsilon = 0.01$, the second quantity is 100× larger than the first one. In this work, we deal with the limit $r \to \infty$ in which the two notions coincide, since all evolutionary trajectories terminate with the mutants fixating.

Third, the fixation time generally depends on the starting location of the mutant. For example, on a large star graph $S_n$ with $r > 1$ fixed, the absorption time of a mutant starting at the center node is roughly $n \log n$ steps, whereas for the mutant starting at any of the leaves it is roughly $n^2 \log n$ steps. One natural approach to handle this is to average over the possible starting positions (either uniformly, or according to some distribution such as the so-called temperature [49]). Alternatively, as we do in this work, one can specify the starting node. Any bounds that hold regardless of the starting node also apply to initializations that average over the starting nodes (including the uniform and temperature initialization).

Fourth, one should specify the units in which the time is measured. This issue is more subtle than it might seem at first glance. In this work, we count steps of a certain slightly modified Moran process (see section Model). However, by far the most popular approach is to count the steps of the classic Moran process [7] and, possibly, in the end normalize by a factor of $n$ to get to "generations" [31]) to capture the fact that the $n$ individuals are reproducing in parallel. The disadvantage of using the steps of the classic Moran process as a basis for measuring time is that it leads to certain counter-intuitive results, see e.g. [35,50].

To present yet another paradoxical consequence of counting the steps of the classic Moran process, consider the population structured as a so-called lollipop graph $L_n$ with $\sqrt{n}$ nodes along a directed path and the remaining nodes in a fully connected cluster (see Fig 4). Biologically, such a structure could represent a stream leading to a pond. If the initial mutant appears at the start of the path, the mutants eventually fixate with probability one – they simply make their way along the path and then they repeatedly invade the cluster until one such attempt succeeds. On average, this happens after a certain number of steps. However, if we initially

**a** Lollipop graph $L_n$                    **b** Comparison of two initializations

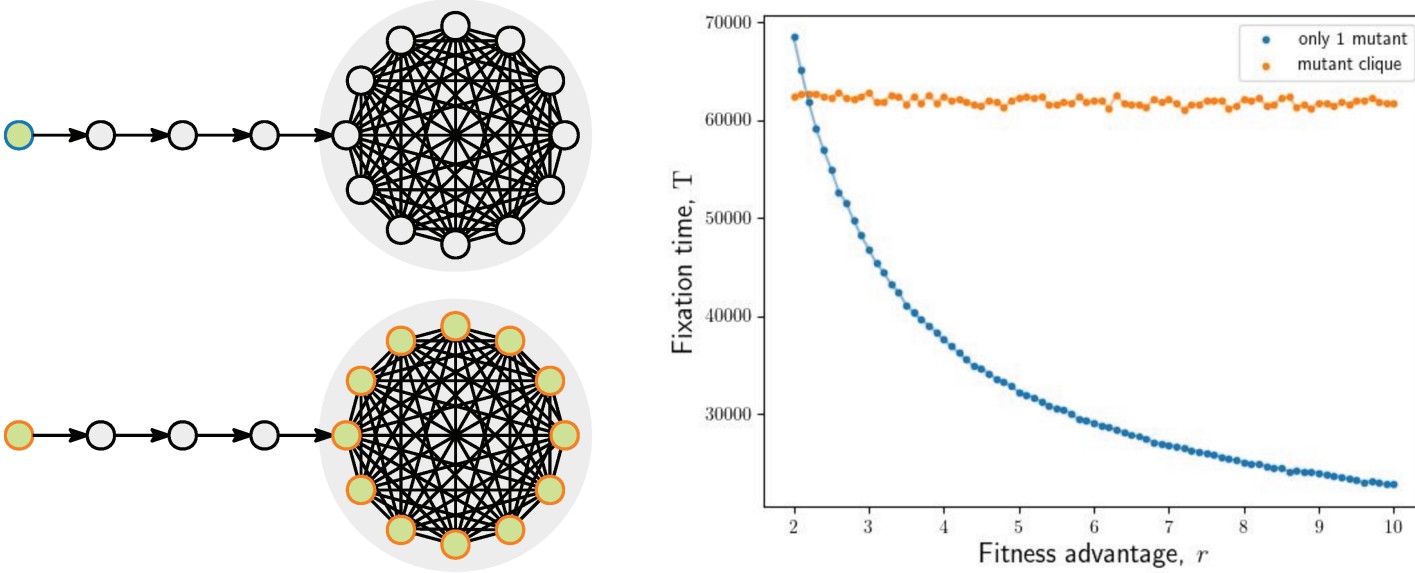

**Fig 4. Starting with more mutants causes more classic Moran steps. a,** The lollipop graph $L_n$ consists of $\sqrt{n}$ nodes arranged along a directed path, and the remaining nodes in a fully connected cluster (here $n = 16$). **b,** The classic fixation time on a lollipop graph $L_n$ with two different initializations: either a single mutant at the start of the path (blue), or additionally all mutants in the fully connected cluster (orange). For $r > 2.2$ the first initialization leads to fewer classic Moran steps, despite having a strict subset of nodes that are initially mutants. Here $n = 1600$, $r \in [2, 10]$, and at least $10^3$ simulations per data point.

place additional mutants in the cluster, then the expected number of steps may increase. Intuitively, this is because mutants in the cluster are selected for reproduction more often than the residents would be, and this slows down the progress of the mutants along the path. This effect becomes especially pronounced in the limit $r \to \infty$. In S1 Text we show that with the first initial condition, the process terminates after roughly $n \log n$ expected Moran steps, whereas with the second one it terminates after roughly $n^{1.5}$ expected Moran steps. Since for large $n$, we have $n^{1.5} \gg n \log n$, we conclude that adding initial mutants might substantially increase the number of steps of the classic Moran process.

To circumvent those paradoxical results, we propose to define the time units in a variable way depending on the total fitness of the population. Formally, if the total fitness of the population is $F$, then we propose that one step of the classic Moran process accounts for $1/F$ units of "real" time. Mathematically, this way of measuring time exactly corresponds to the situation in which Moran process is run in continuous time, and the reproduction time of any individual with fitness $f$ is an exponentially distributed random variable with parameter $1/f$ [51].

In the case of constant selection ($r > 1$ fixed), we have $n \leq F \leq r \cdot n$, therefore each step lasts at least $1/(rn)$ and at most $1/n$ units of real time. Thus, up to a constant factor at most $r$, the real time corresponds to the the standard fixation time measured in generations (rather than in steps).

However, the difference might become much more pronounced in other regimes. For example, consider again the lollipop graph $L_n$ with initially a single mutant at the start of the path (that is, consider the first initial condition from Fig 4). Suppose that mutants have fitness 1 and that residents have fitness 0 (thus we are in the regime $r \to \infty$). In the S1 Text we show that the classic Moran process then takes roughly $n \log n$ steps, which is roughly $\log n$

generations, whereas the real time is roughly $\sqrt{n}$ units. (Intuitively, this is because each of the $\sqrt{n}$ nodes along the directed path must become a mutant, and each one of them becomes a mutant after 1 unit of real time, on average.) Thus, neither classic Moran steps nor generations correctly represent the total duration of the process. On the other hand, we show that the colonization time, which is based on the modified Moran process (see section Model) is roughly $n\sqrt{n}$ steps, which is exactly a factor $n$ more than the real time. In fact, we show that this connection between the colonization time and the real time exists for any graph: in order to compute the real time in the regime where mutants have fitness 1 and residents have fitness 0, one should compute the colonization time (in steps, based on the modified Moran process), and then divide by $n$. This connection is the reason that in the limit $r \to \infty$ we work with the modified Moran process in the first place. See S1 Text for details. To summarize, our bounds on colonization time yield the following bounds on real time: The colonization process terminates after at most $n^2$ units of real time on any graph, after at most $n^{1.5}$ units of real time on any undirected graph, and after at most $n$ units of real time on any regular graph.

## Supporting information

**S1 Text. Formal proofs of the mathematical claims from the main text.**
(PDF)

## Author contributions

**Conceptualization:** Lenka Kopfová, Josef Tkadlec.

**Formal analysis:** Lenka Kopfová, Josef Tkadlec.

**Funding acquisition:** Josef Tkadlec.

**Investigation:** Lenka Kopfová, Josef Tkadlec.

**Methodology:** Lenka Kopfová, Josef Tkadlec.

**Project administration:** Josef Tkadlec.

**Software:** Lenka Kopfová, Josef Tkadlec.

**Supervision:** Josef Tkadlec.

**Visualization:** Lenka Kopfová, Josef Tkadlec.

**Writing – original draft:** Lenka Kopfová, Josef Tkadlec.

**Writing – review & editing:** Lenka Kopfová, Josef Tkadlec.

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
