## [Decision Letter · Decision Letter 0]

13 Jan 2025

PCOMPBIOL-D-24-02021

Colonization times in Moran process on graphs

PLOS Computational Biology

Dear Dr. Tkadlec,

Thank you for submitting your manuscript to PLOS Computational Biology. After careful consideration, we feel that it has merit but does not fully meet PLOS Computational Biology's publication criteria as it currently stands. Therefore, we invite you to submit a revised version of the manuscript that addresses the points raised during the review process.

Please submit your revised manuscript within 30 days Mar 15 2025 11:59PM. If you will need more time than this to complete your revisions, please reply to this message or contact the journal office at ploscompbiol@plos.org. Please include the following items when submitting your revised manuscript:

We look forward to receiving your revised manuscript.

Kind regards,

Mark Broom

Guest Editor

PLOS Computational Biology

Zhaolei Zhang

Section Editor

PLOS Computational Biology

**Additional Editor Comments:**

Three reviewers have considered this interesting paper on a variant of a well established evolutionary process on graphs. All three can see the merits of the work and Reviewers 2 and 3 in particular raise only relatively minor points to address. Reviewer 1 gives a number of points relating to different aspects of the paper, ranging from the generality and relevance of the model to points of detail within the calculations. I believe that this is a strong piece of work and am in broad agreement with the overall positive assessment of the reviewers. The authors should carefully address the points raised by all three reviewers in a revised version of the paper.

**Journal Requirements:**

At this stage, the following Authors/Authors require contributions: Lenka Kopfová, and Josef Tkadlec. Please ensure that the full contributions of each author are acknowledged in the "Add/Edit/Remove Authors" section of our submission form.

5) Your current Financial Disclosure states, "JT: grant PRIMUS/24/SCI/012 from Charles University. The funders had no role in study design, data collection and analysis, decision to publish, or preparation of the manuscript."

However, your funding information on the submission form indicates receiving one fund. Please ensure that the funders and grant numbers match between the Financial Disclosure field and the Funding Information tab in your submission form. Note that the funders must be provided in the same order in both places as well.                                                  . 

Please indicate by return email the full and correct funding information for your study and confirm the order in which funding contributions should appear. Please be sure to indicate whether the funders played any role in the study design, data collection and analysis, decision to publish, or preparation of the manuscript.

**Reviewers' comments:**

Reviewer's Responses to Questions

**Comments to the Authors:**

**Please note that one of the reviews is uploaded as an attachment.**

Reviewer #1: The paper is on evolutionary processes in structured populations that can be represented by graphs. The focus is on the time that is required for mutants’ fixation on graphs for two versions of the Moran process in the extreme case where r→∞. While the paper is well-written and there is nothing wrong with it, the extreme case that is considered by the authors may not be biologically relevant, and hence it may be of less interest. In particular, from an evolutionary point of view, there is actually no competition in the process, resident individuals never reproduce, and mutants inevitably will fixate with probability equal to one given that there is a path from every node to any other node in the graph. The processes in general are oversimplified and some of the results and conclusions are obvious given the nature of the processes that are considered. In many cases, even though valid bounds of the time to fixation are derived, it is not clear that these are tight. The meaning and importance of these bounds is also not clear for a general graph. The approximations on ‘specific graphs’ (e.g. circle, complete, star graphs) may not be of much interest either – exact solutions of the absorption and fixation times have been derived for every fitness scenario, even for scenarios where the fitness advantage of mutants is not constant. On the other hand, while there is indeed a place for such a study in the literature, I am not sure if this is a good fit for PLOS Computational Biology, as there is nothing ‘computational’ in the study.

Some specific comments

Title – I am not sure if the processes that are considered here are evolutionary processes that could be called ‘Moran processes’

Abstract – there is no mention of ‘graphs’, but the focus is on ‘graphs’. The authors mention ‘spatial structures’, but this can mean many different things. When they mention ‘spatial structures with two-way connections’, they may also want to specify that they mean undirected graphs, or explain briefly what they mean by this (also in the ‘Author summary’ and ‘Introduction’).

Author Summary (page 1):

‘We give precise formulas …such as the well-mixed populations, cycles, or stars’ – there are exact formulae for these in the literature for every fitness scenario – The authors could potentially refer to these to verify some of their results in the extreme case where r→∞.

‘We prove a tight general bound’ – I have not been convinced that all the bounds presented are tight – but all are indeed bounds of the times considered.

Introduction

Page 2, ‘migration patterns’ – the authors may want to rephrase, as migration may mean different things, for example real migration of an individual occupying a node, to some other node, leaving the previously occupied node empty.

Page 2, ‘The evolutionary dynamics is governed by the Moran Birth-Death process’ – Many other different processes have also been considered.

Page 2, ‘the number of steps is roughly nlogn’ – is this true for finite populations?

Page 3, ‘In the opposite limit’ – apart from the two extremes, we also have the more natural case where r>1, sometimes it is called as ‘the strong selection case’, where r takes intermediate values. The authors may want to also mention this.

Again, the limit of r→∞, may not describe any natural evolutionary processes.

Page 3, ‘For every population size n we precisely pinpoint the unique population structure with the slowest colonisation time’ – does the slowest structure depends on the size of the population n?

Model

Page 3, ‘lattices of any connectivity’ – edit to ‘of any degree’, as here the authors talk about regular graphs.

Page 4, Colonization time: ‘We will often study the worst-case scenarios’ – the authors should specify what they mean by ‘worst-case scenarios’ – throughout.

Results

Page 5, ‘exactly tight’ – what do the authors mean by this?

Page 6, ‘colonization times range from roughly…’ – what is ‘roughly’? – throughout

Page 6, Under Theorem 3, ‘the colonization times are generally substantially shorter than the fixation times’ – it should be ‘than the fixation times in the case where r>1’, because what the authors call colonization time is eventually the fixation time in the extreme scenario where t->infinity. In addition, I think it is obvious that the ‘colonization times’ will always be lower than that of the the r>1 case (also, ‘Theorem 5’).

Page 6, below Theorem 4: ‘Thus, regular graphs’ should be ‘Thus, regular undirected graphs’?

‘than undirected graphs’ –> ‘than other undirected graphs of the same size’?

Discussion

Page 8: ‘On average, this happens after some number of steps’ – edit.

Supplementary

Page 1, Definition 1 – delete ‘And’ after the first full stop.

Page 1, ‘u for reproduction is selected proportionally’ -> ‘u is selected for reproduction proportionally’

Page 2, ‘a modified version of the classical Moran B-D process’ – I am not sure if this version could be called a version of the Moran process.

Page 2, ‘which captures the idea of a strong selection of mutants’ – is this a natural evolutionary process? There is no competition, and mutants are selected randomly.

In the ‘Classic Moran process’, residents are never chosen ‘for reproduction’. What is the biological relevance of this? On the other hand, in the ‘Continuous process’, residents can be chosen ‘for reproduction’, but they never reproduce, and the steps in which there are chosen are counted in the fixation time. Apart from the mathematical exercise, it would be good if the authors could clarify the biological/evolutionary process that they model. This is not colonisation of an empty environment either.

Page 2, ‘and m spreads into u’ -> ‘and the offspring of m replaces the individual in u’.

The authors refer to the second version of the ‘Moran process’, sometimes as ‘continuous’, sometimes as ‘modified’. It would be good to be consistent. In addition, I would avoid the term ‘continuous’ as this may imply the continuous-time process.

Page 4, Proof of theorem 6:

‘of gaining a mutant’ -> of gaining an active edge?

Double check – is the 1/n in the second summation correct?

Again, the process is oversimplified here, and there are no surprises.

Citations in the Supplementary should be ‘numbered’, as in the main part.

Page 4, Section 2.1: ‘is always faster’ - > ‘is always faster than’

Page 4, Theorem 1/Proof – This is indeed a bound, but not obvious to me that is tight.

Page 5, Theorem 2/Proof: Why this is the slowest?

What is DAGs?

Page 6, Theorem 8/Proof: Not clear to me that this bound is tight? What about the probability of replacement? This is only true in the well-mixed case where the probability of a mutant that is chosen for reproduction replacing a resident is equal to 1 in this particular case. The actual process in each theorem (Classic or modified) should also be specified.

Page 6, ‘of gaining a mutant in stage k’ -> ‘of gaining an active link’ ?

Page 6, Theorem 4 – Figure 4 does not show a regular graph.

Page 6, ‘Or ek is small, but then the new mutant will have many active edges’. Is this still for regular graphs? What about the case of the circle?

Last line of Page 7 – this is just the probability of gaining an active link, and not a new mutant, no?

Page 8, proof of theorem 3: ‘the probability that edge (u,v) is selected for reproduction’ – edit.

Page 8, Proof of theorem 9: It is not clear how tight this bound is given the assumptions, but of course Pk>=1/Dn and E[tk]<=Dn.

Section 5 – I am not sure how much interesting this section may be as long as there are exact formulae for the specific cases presented. The authors could just verify their results with those of previous studies.

Page 9, Theorem 10: ‘Hence, at every time, exactly two active edges connect the ends’ – unless there are n-1 mutants.

Clique vs Complete graph – it would be good to be consistent with the use of the name.

Page 10, Section 5.3: ‘can be obtained by gluing two copies’ ->‘can be obtained by connecting the centre of two copies’

Figure 5 – the circle could also be added here as long as it is considered in this section.

Page 11, Theorem 12: ‘the probability of gaining a new vertex’ -> the probability of gaining a new mutant’?

WLOG – good to write it

Theorem 15: This is only on the ‘lollipop’, no? It should be clarified. Similarly with Theorem 16.

Reviewer #2: Uploaded review as an attachment

Reviewer #3: This paper investigates the population graph structure dependence of colonization times in a simple model of evolutionary dynamics (analogous to an infinite fitness Moran model). The authors present new bounds (supported by rigorous proofs in the SI) on mean colonization times for all graph structures, explicitly construct the worst-case graphs with slowest colonization, and refine these bounds for a variety of special cases of interest. They also provide an insightful discussion regarding timescales of the Moran process and identify counterintuitive nonphysical results that can arise if one interprets Moran steps as real time. Given the generality and rigor of the results, this paper will be a valuable contribution to the evolutionary dynamics literature. The concluding discussion regarding notions of time will be particularly useful for interpreting implications of simplified models (like the Moran model considered here) for real world evolutionary processes. Below I detail several minor clarifying questions about the methods and results; once these are addressed I will fully support publication of this paper.

1. The lower bound shown in Figs. 2,3 should be briefly described in the main text. I would naively expect the complete graph would have fastest colonization, is it obvious whether or not this is true?

2. How would the authors modify the standard Moran update rule for the “continuous process” when the fitness ratio is finite? Can the initialization comparison in Fig. 4 be made for the continuous update rule as well? Is the modified process monotone (in the sense of SI section 6.2)?

3. Do the authors expect the modification of the Moran update rule used in this paper to impact any well-known results in the literature? For example, the authors mention (line 26) the fact that amplifier graphs that increase fixation probability always increase fixation time, is it clear whether this is still true for the modified process?

4. Since this paper focuses only on means, it would be interesting to slightly expand upon the statement (line 224-5) regarding the concentration around the mean fixation time. Besides a few special cases covered in the cited papers, it is not clear to me that the distribution of fixation times is generally well concentrated, particularly for directed graphs. Could the techniques developed in this manuscript be used/generalized to bound higher order moments? What would be the evolutionary implications of large variation in fixation times?

5. For theorem 3 in the SI it would be helpful to better motivate and explain the choice d=sqrt(n). E.g. how does the proof breakdown if another d is chosen to try to tighten the bound? It seems like the choice balances the time contributions from colonizing neighbors of “small” and “large” vertices; is that the correct interpretation?

Minor comments:

— The half path graph, whose colonization time is plotted in Fig. 2 is not mentioned in the main text.

— SI Theorem 5: Should the inequality involving expectation of t_k, t_k^L be reversed?

— SI before Definition 9: acronym DAG should be defined.

**Have the authors made all data and (if applicable) computational code underlying the findings in their manuscript fully available?**

Reviewer #1: Yes

Reviewer #2: Yes

Reviewer #3: Yes

PLOS authors have the option to publish the peer review history of their article (what does this mean?). If published, this will include your full peer review and any attached files.

Reviewer #1: No

Reviewer #2: No

Reviewer #3: No

**Figure resubmission:**
---

## [Editor Report · Decision Letter 1]

10 Feb 2025

Dear Mr. Tkadlec,

We are pleased to inform you that your manuscript 'Colonization times in Moran process on graphs' has been provisionally accepted for publication in PLOS Computational Biology.

Best regards,

Mark Broom

Guest Editor

PLOS Computational Biology

Zhaolei Zhang

Section Editor

PLOS Computational Biology

The authors have given sensible and considered responses to all of the points raised by the reviewers. The reviewers were initially very positive in general, and so I recommend this very nice paper for publication.

---

## [Editor Report · Acceptance letter]

PCOMPBIOL-D-24-02021R1

Colonization times in Moran process on graphs

Dear Dr Tkadlec,

I am pleased to inform you that your manuscript has been formally accepted for publication in PLOS Computational Biology. Your manuscript is now with our production department and you will be notified of the publication date in due course.

With kind regards,

Anita Estes
